# Can Humans Be out of the Loop?

**Junzhe Zhang**                                              JUNZHEZ@CS.COLUMBIA.EDU
**Elias Bareinboim**                                          EB@CS.COLUMBIA.EDU
*Department of Computer Science*
*Columbia University*
*New York, USA*

**Editors:** Bernhard Schölkopf, Caroline Uhler and Kun Zhang

## Abstract

Recent advances in Reinforcement Learning have allowed automated agents (for short, agents) to achieve a high level of performance across a wide range of tasks, which when supplemented with human feedback has led to faster and more robust decision-making. The current literature, in large part, focuses on the human's role during the learning phase: human trainers possess a priori knowledge that could help an agent to accelerate its learning when the environment is not fully known. In this paper, we study an interactive reinforcement learning setting where the agent and the human have different sensory capabilities, disagreeing, therefore, on how they perceive the world (observed states) while sharing the same reward and transition functions. We show that agents are bound to learn sub-optimal policies if they do not take into account human advice, perhaps surprisingly, even when human's decisions are less accurate than their own. We propose the counterfactual agent who proactively considers the intended actions of the human operator, and proves that this strategy dominates standard approaches regarding performance. Finally, we formulate a novel reinforcement learning task maximizing the performance of an autonomous system subject to a budget constraint over the available amount of human advice.

**Keywords:** Causal inference, Graphical models, Reinforcement learning

## 1. Introduction

Sequential decision-making plays a central role in the design of modern intelligent systems. In a prototypical setting, for example, an agent is deployed in an unknown environment and needs to learn how to act such that a set of non-trivial goals are achieved. A solution in these settings usually comes in the form of a policy, namely, a function mapping from the agent's past observations (e.g., visited states, realized outcomes) to actions it should execute at each instant. Human experts might be able to specify a policy in advance for simple tasks, but the policy usually needs to be learned from experimentation in most complex, real-world scenarios. Reinforcement learning (RL) (Sutton and Barto, 1998) has emerged as the *de facto* framework to solve this problem, allowing agents to learn optimal policies by using through interactions with the environment, ideally, without explicit human instructions. One of the major challenges when applying RL methods in practice is the substantial amount of time and data required to find a reasonable, hopefully optimal, policy.

Many efforts have been made to accelerate the learning process by allowing agents to learn by interacting with human trainers. One such approach is to learn from humans past behaviors, which includes *learning from demonstration* (Atkeson and Schaal, 1997; Chernova

and Veloso, 2007; Argall et al., 2009; Hussein et al., 2017; Zhang and Bareinboim, 2017) and *inverse reinforcement learning* Ng and Russell (2000); Abbeel and Ng (2004). In general, these techniques learn a policy by taking as input observations of a human performing a task. Another approach that has shown promise is known as *interactive reinforcement learning*, where the agent is provided in real time with a series of *critiques* from (possibly non-expert) humans (Knox and Stone, 2009, 2012; Mandel et al., 2017). Techniques developed under this rubric include *reward shaping* (Skinner, 1990; Randløv and Alstrøm, 1998; Ng et al., 1999; Devlin and Kudenko, 2012), *policy shaping* (Griffith et al., 2013; Loftin et al., 2014, 2016) and *student-teacher training* (Clouse, 1996; Torrey and Taylor, 2013).

This body of the literature concerned with incorporating the "human in the loop" focuses on the role of human instructors in the learning phase of the system. By and large, the challenge is that an agent has an incomplete model of the environment (e.g., unknown reward functions or system dynamics), which can be improved by human trainers who possess a more detailed understanding about the environment and the task at hand. The translation of the trainer's knowledge into insights for the agent would then allow a more aggressive pruning of the search space, naturally, leading to accelerated learning rates. This approach suggests that after serving as a data source for learning the model, human subjects have no additional value and, therefore, can be disposed of, i.e., moved "out of the loop."

In this paper, we explore a novel aspect of the role of human instructors in RL, specifically after the automated agent learns a model of the environment.[1] We show, by a simple example, that when the agent and the human have different perceptual capabilities (observed states), while sharing the same reward and transition functions, the agent could still be constrained to sub-optimal policies if it does not consider the human's instructions (regardless of the number of interactions allowed, number of demonstrations, etc). Most closely related to our work, (Wray et al., 2016) considered a planning problem where the human and agent repeatedly transfer the control of the decisions, thus reaching a performance level not achievable by the standard autonomous approaches. This paper explicitly accounts for capabilities of the human and agent using the language of causality (Pearl, 2000; Lee and Bareinboim, 2018, 2019; Zhang and Bareinboim, 2019, 2020b; Zhang et al., 2020; Kumor et al., 2021). This allows us to delineate a large number of instances where simple solutions are available, yet overlooked in previous work. More specifically, our contributions are:

1. We demonstrate the conditions under which human input can provide valuable information for agents to achieve optimal performance, even when the model of the environment is fully known.

2. We define the counterfactual agent and identify a class of environments where it can be efficiently optimized using standard MDP algorithms (albeit with changes).

3. We formulate a novel RL problem subject to a budget constraint of available human advice, which is reducible to a constrained MDP. The optimal policy, in this case, is computable as a solution of a polynomial program.

Given the space constraints, all proofs are provided in the complete technical report (Zhang and Bareinboim, 2020a)

---

1. The problem considered here will appear compounded when the agent is also learning system dynamics.

### 1.1. Preliminaries

We introduce in this section the basic notations and definitions used throughout the paper. Variables are denoted by capital letters (e.g., $X$), and their values by lowercase letters ($x$). Let $\mathcal{X}$ denote the domain of a variable $X$. Let $v^{([i,j])}$ stand for a sequence of variables $\left(v^{(i)}, v^{(i+1)}, \ldots, v^{(j)}\right)$ (empty if $j < i$), and $V^{([i,j])} = v^{([i,j])}$ for a sequence of assignments $V^{(i)} = v^{(i)}, V^{(i+1)} = v^{(i+1)}, \ldots, V^{(j)} = v^{(j)}$. We will consistently use the abbreviation $P(x)$ for the probabilities $P(X = x), \forall x \in D(X)$, so does $P(y|x) = P(Y = y|X = x)$. This paper deals extensively with finite (PO)MDPs models as the basis for designing RL algorithms.

**Definition 1 (Partially Observed MDP)** *A finite POMDP (Cassandra et al., 1994) is a 6-tuple $\langle \mathcal{S}, \mathcal{X}, \mathcal{O}, T, R, \Omega \rangle$ in which $\mathcal{S}$ is a finite set of states; $\mathcal{X}$ a finite set of actions; $\mathcal{O}$ is a finite set of observations; $T : \mathcal{S} \times \mathcal{X} \times \mathcal{S} \to [0,1]$ is a transition distribution; $R : \mathcal{S} \times \mathcal{X} \to \mathbb{R}$ is a reward function; and $\Omega : \mathcal{S} \times \mathcal{O} \to [0,1]$ is an observation function.*

Formally, if states are fully observed (i.e., $\mathcal{O} = \mathcal{S}$ and $\Omega(s, x, s) = 1$), the 4-tuple $\langle \mathcal{S}, \mathcal{X}, T, R \rangle$ forms an MDP. For simplicity, let $T_x^s(s') = T(s, x, s')$, $R_x^s = R(s, x)$ and $\Omega^s(o) = \Omega(s, o)$.

## 2. The Value of Human Advice

We start the discussion with an example involving the sequential decision-making of patients' treatment. We show that an agent with the fully specified system dynamics could fail to find an effective policy if it ignores the advice of the human instructor.

Specifically, a physician treats each patient who visits the hospital regularly to maintain her long term health condition. This sequential environment can be represented by an MDP $M_{\text{MDPUC}}$ described in Fig. 2(a); here, UC stands for unobserved confounders. More specifically, the physician measures the patient's corticosteroid level at the $t$-th visit, $S^{(t)} = s^{(t)} \in \{0, 1\}$, where 0 stands for a low and 1 for a high

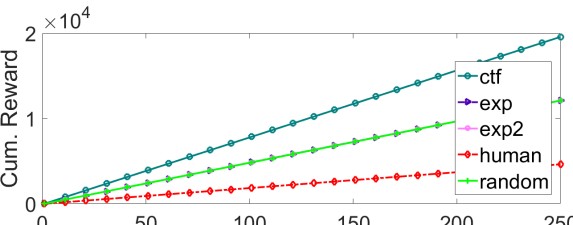

Figure 1: Simulations comparing experimental and counterfactual agents.

level of corticosteroid. She then decides a treatment $X^{(t)} = x'^{(t)} \in \{0, 1\}$ (1 for to give the drug, 0 for not to), and then measures an overall health score $Y^{(t)} = y^{(t)} \in \{0, 1\}$ (i.e., 1 for "healthy" and 0 for "not healthy"). In reality, the patient's health score $Y^{(t)}$ is also affected by a pair of variables $U^{(t)} = \{M^{(t)}, E^{(t)}\}$, where $M^{(t)} = m^{(t)}$ stands for patient's psychological status (0 for a positive mood, 1 for negative) and $E^{(t)} = e^{(t)}$ stands for her socioeconomic status (0 for wealthy, 1 for poor). The patient's long-term health condition can be modeled by the long-term cumulative reward $\sum_{t=1}^{\infty} \gamma^t Y^{(t)}$, $\gamma = 0.99$ The physician decides a treatment $X^{(t)} = x'^{(t)}$ based on values of $S^{(t)}, M^{(t)}, E^{(t)}$, which we summarize as a policy function $x'^{(t)} = \pi_h\left(s^{(t)}, m^{(t)}, e^{(t)}\right)$. The detailed parametrizations of the transition function $T$, the reward function $R$, and the policy $\pi_h$ are described in Appendix A.

To maximize the patient's long-term health, the hospital's administration aims to automate the decision procedure and deploy an autonomous agent $A_{\text{EXP}}$. The agent learns the fully specified model (functions $T$ and $R$) from the physician, but it doesn't have access

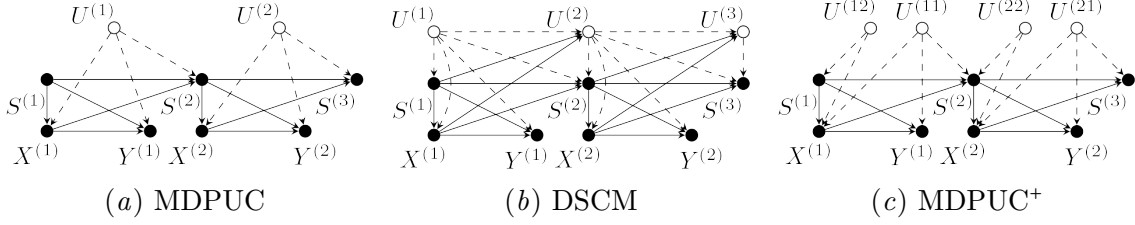

Figure 2: Causal diagrams for different environmental models: (a) MDPUC where $U^{(t)}$ affects only $\{X^{(t)}, Y^{(t)}, S^{(t+1)}\}$; (b) a general DSCM (Thm. 2); (c) MDPUC$^+$ where $U^{(t)} = \{U^{t1}, U^{t2}\}$ affects $\{X^{(t)}, Y^{(t)}, S^{(t+1)}\}$, and also $S^{(t)}$.

to the psychological mood $M^{(t)}$ and the socioeconomic status $E^{(t)}$ of the patient due to privacy concerns. That is, $M^{(t)}, E^{(t)}$ are unobserved variables for the learning agent. We solve for this system using both the standard MDP and POMDP planning algorithms and label the resulting policies as *exp* and *exp2*, respectively. The administration also considers a simple agent ($A_{\mathrm{CTF}}$), which observes the corticosteroid level $S^{(t)}$ and also take the physician's decision as advice in real-time (labeled as *ctf*). Two baseline policies are included for comparison: (1) the physician's current policy (called *human*), and (2) a policy picking the treatment at random (*random*). The cumulative reward of all policies are shown in Fig. 1. Perhaps surprisingly, the performance of *exp* and *exp2* coincides with the *random* policy, indicating that $A_{\mathrm{EXP}}$ is unable to learn a reasonable policy, which should be better than chance. Moreover, the interactive approach (*ctf*) manages to outperform all the other policies, despite the fact that the physician (*human*) performs worse than random guessing.

Puzzling questions arise at this point – first, how could the human advice be useful for the performance of the agent even when it has full knowledge of the reward function $R$ and the world dynamics $T$? Second, how could we formalize the interactive agent in this novel setting and systematically find its optimal policy? Our goal for the remainder of the paper is to answer these fundamental questions in full generality (i.e., for any decision setting).

## 3. Modeling Autonomous Systems

Our analysis relies on the semantical framework of structural causal models (SCMs) (Pearl, 2000) . We will describe sequential decision-making systems using dynamic SCMs, which describes causal dynamics of how the environment's states evolve over time.

**Definition 2 (Dynamic SCMs)** *A dynamic structural causal model (DSCM) M is a pair of SCMs $\langle M_1, M_\rightarrow \rangle$ (Pearl, 2000, Ch. 7) with domains over actions $\mathcal{X}$, observed states $\mathcal{S}$, unobserved states $\mathcal{U}$, and rewards $\mathcal{Y}$. $M_1$ is a SCM over initial states $X^{(1)}, U^{(1)}$, actions $X^{(1)}$ and rewards $Y^{(1)}$. $M_\rightarrow$ is a SCM describing the process: for $t = 2, 3, \ldots,$*

*1. $U^{(t)}$ is the unobserved state whose values $u^{(t)} \in \mathcal{U}$ is decided by a function $u^{(t)} = f_u\left(x^{(t-1)}, s^{(t-1)}, u^{(t-1)}, \epsilon_u^{(t)}\right)$, where $\mathcal{E}_u^{(t)}$ is an independent noise drawn from $P(\epsilon_u)$[2];*

---

2. For simplicity, we ignore $\epsilon_u^{(t)}$ and write $u^{(t)} = f_u\left(x^{(t-1)}, s^{(t-1)}, u^{(t-1)}\right)$. The same applies to $f_s, f_y, \pi_h$.

2. $S^{(t)}$ is the observed state whose values $s^{(t)} \in \mathcal{S}$ is decided by a transition function $s^{(t)} = f_s\left(x^{(t-1)}, s^{(t-1)}, u^{([t-1,t])}\right)$;

3. $Y^{(t)}$ is the observed reward whose values $y^{(t)} \in \mathcal{Y}$ is decided by a reward function $y^{(t)} = f_y\left(x^{(t)}, s^{(t)}, u^{(t)}\right)$;

4. $X^{(t)}$ is the action of the human instructor where its values $x^{(t)} \in X$. $x^{(t)}$ is decided by a policy function $x^{(t)} = \pi_h\left(s^{(t)}, u^{(t)}\right)$.

The definition of DSCM is a direct translation of POMDP formalism to causal language: the pair $S^{(t)}, U^{(t)}$ represents the underlying states, and $S^{(t)}$ for the partial observation[3]. Each DSCM $M$ is associated with a causal diagram $G$, which is a directed acyclic graph (DAG) where empty nodes correspond to unobserved variables, solid nodes correspond to observed variables, and edges represent functional (cause-and-effect) relationships (see Fig. 2).

We next model the standard autonomous agents as the experimental agents, which interact with the environment through a series of causal interventions, oblivious to the presence of the human instructor.

**Definition 3 (Experimental Agent)** *An experimental agent $A_{\mathrm{EXP}}$ is defined as a sequence of policies $\Pi_{\mathrm{EXP}} = \pi^{([1,t])}$. Each $\pi^{(t)}$ is a function deciding values of $X^{(t)}$ such that $x^{(t)} = \pi^{(t)}\left(h_{\mathrm{EXP}}^{(t)}\right)$, where the experimental history $h_{\mathrm{EXP}}^{(t)} = \left\{s^{([1,t])}, x^{([1,t-1])}\right\}$ is the observed history of $A_{\mathrm{EXP}}$ up to time $t$.*

Using Def. 3, we could model the interaction between an autonomous agent $A_{\mathrm{EXP}}$ and the environment $M_{\mathrm{DSCM}}$ as an intervention on actions nodes $X^{([1,t])}$, denoted by $do\left(\Pi_{\mathrm{EXP}}\right)$, which sets values of $X^{(t)}$ to $\pi^{(t)}\left(h_{\mathrm{EXP}}^{(t)}\right)$ regardless of how they were ordinarily determined by the human operator $\pi_h$ (Pearl, 2000, Ch. 7). To understand the decision process encoded here, note that each step $t$, the agent starts from a state $S^{(t)} = s^{(t)}, U^{(t)} = u^{(t)}$, performs an intervention $X^{(t)} = \pi\left(s^{(t)}\right)$, receives a reward $Y^{(t)} = y^{(t)}$, and observes $S^{(t+1)} = s^{(t+1)}$. We use $Y_{x^{([1,t])}}\left(u^{([1,t])}\right)$ to denote the response of a variable $Y$ to a sequence of interventions $X^{(1)} = x^{(1)}, \ldots, X^{(t)} = x^{(t)}$. This stochastic process is summarized using the counterfactual distribution $P\left(Y_{x^{(t)}}^{(t)} = y^{(t)} \mid S^{(t)} = s^{(t)}, U^{(t)} = u^{(t)}\right)$, abbreviated as $P\left(y_{x^{(t)}}^{(t)} \mid s^{(t)}, u^{(t)}\right)$; so does $P\left(s_{x^{(t)}}^{(t+1)} \mid s^{(t)}, u^{(t)}\right)$. We assess the performance of an agent $\Pi$ using the cumulative rewards $V^{\Pi} = \mathbb{E}\left[\sum_{t=0}^{\infty} \gamma^t Y_{x^{(t)}=\pi^{(t)}}^{(t)}\right]$ where $\gamma \in [0,1)$ is a discount factor.

The Markov property delineates a set of decision-making systems that could be efficiently solved. We next define the Markov property associated with the experimental agents.

**Definition 4 (Experimental Markov Property)** *An environment $M$ is said to be Experimentally Markov (exp-Markov, for short) if and only if for $t \geq 1$,*

$$P\left(s_{x([1,t])}^{(t+1)} \mid s_{x([1,t-1])}^{([1,t])}\right) = P\left(s_{x^{(t)}}^{(t+1)} \mid s^{(t)}\right), \qquad E\left[Y_{x([1,t])}^{(t)} \mid s_{x([1,t-1])}^{([1,t])}\right] = E\left[Y_{x^{(t)}}^{(t)} \mid s^{(t)}\right], \qquad (1)$$

*where $s_{x([1,t-1])}^{([1,t])} = \left\{s_{x([1,t-1])}^{(t)}, s_{x([1,t-2])}^{(t-1)}, \ldots, s^{(1)}\right\}$.*

---

3. DSCMs are defined from an agent's perspective. Unobserved states $U^{(t)}$, $t = 1, 2, \ldots$ could be observable to the human, and vice versa.

In Def. 4, $S^{(t)}_{x^{([1,t-1])}}$ is the potential response of the state $S^{(t)}$ to intervention $do\left(x^{([1,t-1])}\right)$, which can be read as "The value that the state $S^{(t)}$ would be had $X^{([1,t-1])}$ been $x^{([1,t-1])}$" (Halpern, 1998; Pearl, 2000). The exp-Markov property says that for an experimental agent, the information in $S^{([1,t])}$ and the causal influence of the intervention $do\left(x^{([1,t])}\right)$ can be best summarized in the current state $S^{(t)}$ and intervention $do\left(x^{(t)}\right)$. For example, the MDPUC$^+$ model of Fig. 2(c) is exp-Markov though the unobserved states $U^{(t)} = \left\{U^{(t1)}, U^{(t2)}\right\}$ could affect the states $S^{(t-1)}, S^{(t)}$. Consequently, for an exp-Markov environment $M$, the resulting autonomous system $\langle M, A_{\text{EXP}}\rangle$ forms an MDP. The optimal policy $\Pi$ for an MDP is stationary where every policy function $\pi^{(t)}$ only depends on the state $s^{(t)}$, $t = 1, 2, \ldots$, and is invariant to time $t$, which we denote by $\pi$.

Armed with these definitions, we are finally ready to analyze the medical treatment example and see how the human advice provides critical information to the task. First note that MDPUC$^+$ is a generalization of the MDPUC model of Fig. 2(a), the environment $M_{\text{MDPUC}}$ is exp-Markov. The optimal policy of the autonomous agent $A_{\text{EXP}}$ is stationary and takes the form of $x^{(t)} = \pi\left(s^{(t)}\right)$. On the other hand, the physician follows a policy $\pi_h$ that takes as input the values $s^{(t)}, m^{(t)}, e^{(t)}$. The difference in the domains of the policies $\pi$ and $\pi_h$ represents different capabilities of the agent and the human. In this case, the physician possesses superior observation capacities as she could access the states $M^{(t)}, E^{(t)}$, which are unavailable for the agent. The autonomous agent carries a series of interventions $do\left(\pi\right)$, which replaces the physician's policy $\pi_h$ with its policy $\pi$. Since $M^{(t)}, E^{(t)}$ affect the values of reward $Y^{(t)}$ and the next state $S^{(t+1)}$, the physician's decision $x'^{(t)}$ could contain critical information for the long-term health condition of the patients. Discarding the physician's decision $x'^{(t)}$ thus leads to potentially sub-optimal policies.

## 4. Counterfactual Agents

We will account for the human interaction in the environment using the language of SCMs. A counterfactual agent takes the human's advice as its intended action, and adjusts this action using the counterfactual reasoning if a better decision is available (the intended action is called *intuition* in (Bareinboim et al., 2015; Forney et al., 2017).

**Definition 5 (Counterfactual Agent)** *A counterfactual agent $A_{\text{CTF}}$ is defined as a policy $\Pi_{\text{CTF}} = \pi^{([1,t])}$. $\pi^{(t)}$ is a function deciding values of $X^{(t)}$ such that $x^{(t)} = \pi^{(t)}\left(h^{(t)}_{\text{CTF}}\right)$, where the counterfactual history $h^{(t)}_{\text{CTF}} = \left\{s^{([1,t])}, x^{([1,t-1])}, x'^{([1,t])}\right\}$, and $x'^{([1,t])}$ is a sequence of human's decisions up to time $t$.*

For an agent $A_{\text{CTF}}$, the evaluation of the policy $\Pi_{\text{CTF}}$ is indeed a counterfactual reasoning procedure: "Given that I am about to perform $x'^{(t)}$ (intended, factual) and the past history, what the world state and reward would be had I done the action $x^{(t)}$ (contrary to the fact, counter-factual)". For concreteness, consider a counterfactual agent $A_{\text{CTF}}$ in the MDPUC model with the goal of maximizing the reward at $t = 1$. Given the observed state $s^{(1)}$ and its intended action $x'^{(1)}$, the agent $A_{\text{CTF}}$ decides for a new action $x^{(1)}$ to maximize the expected reward $Y^{(1)}$, i.e., the counterfactual statement $E\left[Y^{(1)}_{x^{(1)}} \mid s^{(1)}, x'^{(1)}\right]$, which can be read as "The value that the reward $Y^{(1)}$ would attain had I done $x^{(1)}$, given that I am about

to do $x'^{(1)}$ and the current state is $s^{(1)}$." Note that the values of $x'^{(1)}$ (factual) and $x^{(1)}$ (counterfactual) could be different. It follows immediately after Def. 5 that a counterfactual agent consistently dominates an experimental agent in terms of the performance.

**Corollary 6** *Let the optimal policy for $A_{\text{EXP}}$ denoted by $\Pi^*_{\text{EXP}}$ and for $A_{\text{CTF}}$ by $\Pi^*_{\text{CTF}}$. For any DSCM model, $V^{\Pi^*_{\text{EXP}}} \leq V^{\Pi^*_{\text{CTF}}}$.*

Given the space constraints, all proofs are included in Appendix B. Corol. 6 reassures the intuition that, since the counterfactual history always covers the experimental one (i.e., $h^{(t)}_{\text{EXP}} \subseteq h^{(t)}_{\text{CTF}}$), counterfactual agents are never worse than their experimental counterparts. The following result provides a graphical condition for when the performance of $A_{\text{CTF}}$ and $A_{\text{EXP}}$ coincides, i.e., the human advice can be discarded without loss of information.

**Theorem 7** *For any DSCM model where the human decision is affected only by observable variables (arrow $U^{(t)} \to X^{(t)}$ is not present), denoted by $DSCM^-$, $V^{\Pi^*_{\text{EXP}}} = V^{\Pi^*_{\text{CTF}}}$.*

In words, Corol. 6 and Thm. 7 challenge the somewhat popular belief that the role of human instructors in RL systems is to serve as data sources for agents with partially specified models, and it is dispensable once the learning procedure is complete. These results imply that (1) it is still better to utilize an interactive approach ($A_{\text{CTF}}$) even if the information affecting the human decision has an adverse effect on its performance; (2) the human advice is indispensable if it possesses some information about latent states (arrow $U^{(t)} \to X^{(t)}$ is present), even when the agent has obtained the fully specified world model.

### 4.1. Optimizing Counterfactual Agents

We will next discuss methods to systematically compute the optimal policy of counterfactual agents. We first characterize the Markov property associated with the counterfactual agent, as it outlines sufficient statistics for $A_{\text{CTF}}$.

**Definition 8 (Counterfactual Markov Property)** *An environment $M$ is said to be counterfactually Markov (ctf-Markov, for short) if and only if for $t \geq 1$,*

$$P\left(s^{(t+1)}_{x([1,t])}, x'^{(t+1)}_{x([1,t])} \mid s^{([1,t])}_{x([1,t-1])}, x'^{([1,t])}_{x([1,t-1])}\right) = P\left(s^{(t+1)}_{x^{(t)}}, x'^{(t+1)}_{x^{(t)}} \mid s^{(t)}, x'^{(t)}\right), \tag{2}$$

$$E\left[Y^{(t)}_{x([1,t])} \mid s^{([1,t])}_{x([1,t-1])}, x'^{([1,t])}_{x([1,t-1])}\right] = E\left[Y^{(t)}_{x^{(t)}} \mid s^{(t)}, x'^{(t)}\right], \tag{3}$$

*where $x'^{([1,t])}_{x([1,t-1])} = \left\{x'^{(t)}_{x([1,t-1])}, x'^{(t-1)}_{x([1,t-2])}, \ldots, x'^{(1)}\right\}$ and $s^{([1,t])}_{x([1,t-1])}$ follows Def. 4.*

Compared to the exp-Markov property (Def. 4), Eqs. (2) and (3) condition not only on the observed states $s^{([1,t])}_{x([1,t-1])}$ but all the past human's decisions $x'^{([1,t])}$ to interventions $do\left(x^{([1,t-1])}\right)$ as well. This additional term makes the independence conditions for ctf-Markov property somewhat more involved: models that are exp-Markov are not necessarily ctf-Markov, as shown in the next example.

**Proposition 9** *$M_{\text{MDPUC}^+}$ (Fig. 2(c)) is not ctf-Markov.*

Prop. 9 confirms that, in general, ctf-Markov property does not necessarily hold in a exp-Markov environment. However, the ctf-Markov property is satisfiable in some settings, which is characterized by the general graphical condition given in the next result.

**Theorem 10** $M_{\mathrm{MDPUC}}$ *(Fig. [2](b)) is ctf-Markov.*

For a ctf-Markov environment $M_{\mathrm{MDPUC}}$, the interactive RL system $\langle M, A_{\mathrm{CTF}} \rangle$ forms an MDP where the human decision is included as an additional observed state. More formally, the MDP should be constructed as follows:

**Proposition 11** $\langle M_{\mathrm{MDPUC}}, A_{\mathrm{CTF}} \rangle$ *forms an MDP* $\langle \mathcal{S}_{\mathrm{CTF}}, \mathcal{X}_{\mathrm{CTF}}, T, R \rangle$, *where* $\mathcal{S}_{\mathrm{CTF}} = \mathcal{S} \times \mathcal{X}$, $\mathcal{X}_{\mathrm{CTF}} = \mathcal{X}$, *and for any* $x, s, x', x''$,

$$T_x^{\langle s, x' \rangle}\left(s', x''\right) = P\left(S_{X^{(t)}=x}^{(t+1)} = s', X_{X^{(t)}=x}^{(t+1)} = x'' \mid S^{(t)} = s, X^{(t)} = x'\right), \tag{4}$$

$$R_x^{\langle s, x' \rangle} = E\left[Y_{X^{(t)}=x}^{(t)} \mid S^{(t)} = s, X^{(t)} = x'\right]. \tag{5}$$

Prop. 11 follows immediately after the ctf-Markov property (Thm. 10). The transition $T$ and the reward function $R$ are both counterfactual quantities, which can be computed by the procedure provided in (Pearl, 2000, Thm. 7.1.7), whenever the SCM is known. Alternatively, they can be empirically estimated using the new counterfactual randomization procedure introduced in Bareinboim et al. (2015).

There are several standard ways of solving an MDP with the discounted expected cumulative rewards optimization criterion Puterman (2014). Some use dynamic programming (value of policy iteration), others reduce MDPs to linear programs (LPs). A discounted MDP formed by an interactive RL system $\langle M_{\mathrm{MDPUC}}, A_{\mathrm{CTF}} \rangle$ can be formulated as the following LP d'Epenoux (1963); Kallenberg (1983) (this maximization LP is the dual to the more commonly used minimization LP in the value function coordinates):

$$\max \sum_{s,x,x'} R_x^{\langle s, x' \rangle} \phi_x(s, x') \tag{6}$$

$$\text{subject to } \phi_x(s, x') \geq 0$$

$$\sum_x \phi_x(s, x') - \gamma \sum_{i,x} \phi_x(i) T_x^i(s, x') = \alpha_{\langle s, x' \rangle}$$

where $\alpha_{\langle s, x' \rangle} = P(S^{(1)} = s, X^{(1)} = x')$ specify the probility distribution over the initial state. The optimization variables $\phi_x(s, x')$ are called the *occupation measure* of a policy, where $\phi_x(s, x')$ is the total discounted number of times action $x$ is executed in state $s, x'$. An optimial policy for $A_{\mathrm{CTF}}$ is stationary and can be computed from a solution to the above LP as $\pi_{\mathrm{CTF}}^*(x|s, x') = \phi_x(s, x') / \sum_x \phi_x(s, x')$.

For a general environment $M_{\mathrm{DSCM}}$, it immediately follows from Prop. 9 that $M_{\mathrm{DSCM}}$ is not ctf-Markov, since DSCM is a generalization of MDPUC$^+$. To solve for the counterfactual agent in an environment where ctf-Markov doesn't hold, a simple realization is to resort to POMDPs by using the human's decision $X^{(t)} = x'^{(t)}$ as an additional observation.

**Proposition 12** *A RL system* $\langle M_{\mathrm{DSCM}}, A_{\mathrm{CTF}} \rangle$ *forms a POMDP* $\langle \mathcal{S}_{\mathrm{CTF}}, \mathcal{X}_{\mathrm{CTF}}, \mathcal{O}_{\mathrm{CTF}}, T, R, \Omega \rangle$ *where* $\mathcal{S}_{\mathrm{CTF}} = \mathcal{S} \times \mathcal{U}$, $\mathcal{X}_{\mathrm{CTF}} = \mathcal{X}$, $\mathcal{O}_{\mathrm{CTF}} = \mathcal{S} \times \mathcal{X}$, *and for any* $x, s, u, x', s', u'$,

$$T_x^{\langle s, u \rangle}\left(s', u'\right) = P\left(S_{X^{(t)}=x}^{(t+1)} = s', U_{X^{(t)}=x}^{(t+1)} = u' \mid S^{(t)} = s, U^{(t)} = u\right),$$

$$R_x^{\langle s, u \rangle} = E\left[Y_{X^{(t)}=x}^{(t)} \mid S^{(t)} = s, U^{(t)} = u\right],$$

$$\Omega^{s, u}\left(s', x'\right) = P\left(S^{(t+1)} = s', X^{(t+1)} = x' \mid S^{(t+1)} = s, U^{(t+1)} = u\right).$$

Since the underlying state is not perfectly observable, the agent can maintain a belief $B^{(t)}$ about the current state $(s^{(t)}, u^{(t)})$ at each time step $t$, defined as

$$B^{(t)}\left(s^{(t)}, u^{(t)}\right) = P\left(s^{(t)}_{x([1,t-1])}, u^{(t)}_{x([1,t-1])} \mid s^{([1,t])}_{x([1,t-1])}, x'^{([1,t])}_{x([1,t-1])}\right)$$

The belief can be updated as times progresses based on Bayes' theorem. Formally, the belief update step for $t+1$ goes as follows:

**Theorem 13** *Given the belief $B^{(t)}$, action $x^{(t)}$, next observed state $s^{(t+1)}$, and next human's decision $x'^{(t+1)}$, the next belief can be updated as:*

$$\begin{aligned}
B^{(t+1)}\left(s^{(t+1)}, u^{(t+1)}\right) &= \alpha P\left(s^{(t+1)}, x'^{(t+1)} \mid s^{(t+1)}, u^{(t+1)}\right) \\
&\cdot \sum_{s^{(t)}} \sum_{u^{(t)}} P\left(s^{(t+1)}_{x^{(t)}} u^{(t+1)}_{x^{(t)}} \mid s^{(t)}, u^{(t)}\right) B^{(t)}\left(s^{(t)}, u^{(t)}\right)
\end{aligned} \tag{7}$$

*where $\alpha$ is a normalizing constant.*

Since the belief state is a sufficient statistic, the optimal policy of $A_{\text{CTF}}$ is then the solution of a "belief state MDP" over continous space. There are several standard methods for solving such MDPs, including witness algorithm (Littman, 1994), linear support algorithm (Cheng, 1988), and incremental pruning (Cassandra et al., 1997), just to cite a few.

## 5. The Trade-Off between Autonomy and Optimality

We start this section by summarizing in Table 1 the results discussed so far. The "optimality" column indicates whether the corresponding agent could obtain the optimal policy (using the counterfactual agent $A_{\text{CTF}}$ as the baseline); a check (cross) mark in $A_{\text{EXP}}$ under "optimality" represents that the experimental agent is (is not) able to achieve the optimal performance in the environment of the corresponding row, compared to its counterfactual counterpart. In an arbitrary environment, full autonomy (represented by the "autonomy" column)

| Environment | Optimality | | Autonomy |
| --- | --- | --- | --- |
| | $A_{\text{EXP}}$ | $A_{\text{CTF}}$ | |
| MDPUC⁻ | ✓ | ✓ | ✓ |
| MDPUC | ✗ | ✓ | ✗ |
| MDPUC⁺ | ✗ | ✓ | ✗ |
| DSCM⁻ | ✓ | ✓ | ✓ |
| DSCM | ✗ | ✓ | ✗ |

Table 1: The performance of counterfactual agents $A_{\text{EXP}}$ and experimental agents $A_{\text{CTF}}$ in canonical environments.

can only be achieved when it is not at the cost of the optimality, i.e., the performance of the experimental $A_{\text{EXP}}$ and counterfactual agent $A_{\text{CTF}}$ coincides. These results conveys an inherent trade-off between optimality and autonomy when designing RL systems – while full autonomy is preferable, the agent could potentially achieve better performance by leveraging the human's capabilities. We model this trade-off as a constrained transfer of control (TOC) problem such that a system tries to maximize its rewards while repeatedly transferring between an experimental and a counterfactual agent, subject to the total time of the counterfactual agent (using human decision) is no more than $\delta$ ratio of the total running time. To solve this problem, we first define hybrid agents, which combines both the experimental $A_{\text{EXP}}$ and counterfactual $A_{\text{CTF}}$ modes of reasoning.

**Definition 14 (Hybrid Agent)** *A hybrid agent $A_{\text{HYB}}$ is an agent following a policy $\Pi_{\text{HYB}} = (\pi \circ f_a)^{([1,t])}$ and maintaining an internal state $A^{(t)} \in \{a_0, a_1\}$ ($a_0$ for $A_{\text{EXP}}$ and $a_1$ for $A_{\text{CTF}}$). $(\pi \circ f_a)^{(t)}$ is a composite function in the form of $\pi^{(t)}\left(h_{\text{HYB}}^{(t)}, x'^{(t)}, f_a^{(t)}\left(h_{\text{HYB}}^{(t)}\right)\right)$, where $h_{\text{HYB}}^{(t)}$ is the history of the hybrid agent (includes $s^{([1,t])}$, $x^{([1,t])}$, and partially $x'^{([1,t])}$), the function $a^{(t)} = f_a^{(t)}\left(h_{\text{HYB}}^{(t)}\right)$ decides a value of $A^{(t)}$ and $\pi^{(t)}$ decides the decision $x^{(t)}$ as follows:*

$$\pi^{(t)}\left(h_{\text{HYB}}^{(t)}, x'^{(t)}, a^{(t)}\right) = \begin{cases} \pi_0^{(t)}\left(h_{\text{HYB}}^{(t)}\right), & \text{if } a^{(t)} = a_0 \\ \pi_1^{(t)}\left(h_{\text{HYB}}^{(t)}, x'^{(t)}\right), & \text{if } a^{(t)} = a_1 \end{cases} \tag{8}$$

A hybrid RL system is defined as the pair $\langle M_{\text{DSCM}}, A_{\text{HYB}} \rangle$ proactively deciding whether it considers the human decision. The internal state $A^{(t)}$ is a switch variable indicating the current running mode the system is in ($a_0$ for experimental or $a_1$ for counterfactual). The human decision $x'^{(t)}$ is included as an evidence to decide for agent's decision $x^{(t)}$ if $A^{(t)} = a_1$, and ignored otherwise. We now consider algorithms to solve for such hybrid systems when the environment is ctf-Markov and full autonomy cannot be achieved (e.g., $M_{\text{MDPUC}}$). The following result establishes the relation between hybrid systems and MDPs.

**Proposition 15** *A hybrid system $\langle M_{\text{MDPUC}}, A_{\text{HYB}} \rangle$ forms an MDP $\langle \mathcal{S}_{\text{HYB}}, \mathcal{X}_{\text{HYB}}, T, R \rangle$ where $\mathcal{S}_{\text{HYB}} = \mathcal{S} \times \mathcal{X}$, $\mathcal{X}_{\text{HYB}} = \mathcal{X} \times \mathcal{A}$, and for any $a, x, s, x', s', x''$,*

$$T_{\langle x,a \rangle}^{\langle s,x' \rangle}(s', x') = P\left(S_{X^{(t)}=x}^{(t+1)} = s', X_{X^{(t)}=x}^{(t+1)} = x'' \mid S^{(t)} = s, X^{(t)} = x'\right),$$
$$R_{\langle x,a \rangle}^{\langle s,x' \rangle} = E\left[Y_{X^{(t)}=x}^{(t)} \mid S^{(t)} = s, X^{(t)} = x'\right].$$

This result follows immediately after Prop. 11, as $A^{(t)}$ is an auxiliary action which does not directly affect the environment. Prop. 15 allows us to translate the constrained TOC problem to a constrained MDP problem. The optimal policy of $A_{\text{HYB}}$ can be computed as a solution to the LP specified in Eq. (6) with additional constraints as follows:

$$\sum_{i,x} \phi_{\langle x,a_1 \rangle}(i) \leq \delta \sum_{i,x,a} \phi_{\langle x,a \rangle}(i) \tag{9}$$

$$\phi_{\langle x,a_0 \rangle}(s, x') / \sum_{x} \phi_{\langle x,a_0 \rangle}(s, x') = \sum_{x'} \phi_{\langle x,a_0 \rangle}(s, x') / \sum_{x,x'} \phi_{\langle x,a_0 \rangle}(s, x') \tag{10}$$

$$\sum_{x} \phi_{\langle x,a \rangle}(s, x') / \sum_{x,a} \phi_{\langle x,a \rangle}(s, x') = \sum_{x,x'} \phi_{\langle x,a \rangle}(s, x') / \sum_{x,a,x'} \phi_{\langle x,a \rangle}(s, x') \tag{11}$$

Eq. (9) ensures that for system $\langle M_{\text{MDPUC}}, A_{\text{HYB}} \rangle$, the total time running in counterfactual mode ($a_1$) is no more than $\delta$ ratio of the total running time (discounted so that future visits count less than present ones). Eq. (10) ensures that for $A^{(t)} = a_0$, the policy $\pi$ does not take the human decision $x'^{(t)}$ as an input.Eq. (11) reflects the functional constraint that $f_a$ is a Markovian policy only depends on the current state $s^{(t)}$. This mathematical program forms a polynomial optimization problem (Tuy et al., 1998), which is neither linear nor convex (due to Eqs. (10) to (11)). Despite its difficulty, there are several efficient methods of polynomial optimization that can be used in this case, for example: the RLT method (Sherali and Adams, 2013), and a SDP relaxation method (Lasserre, 2001).

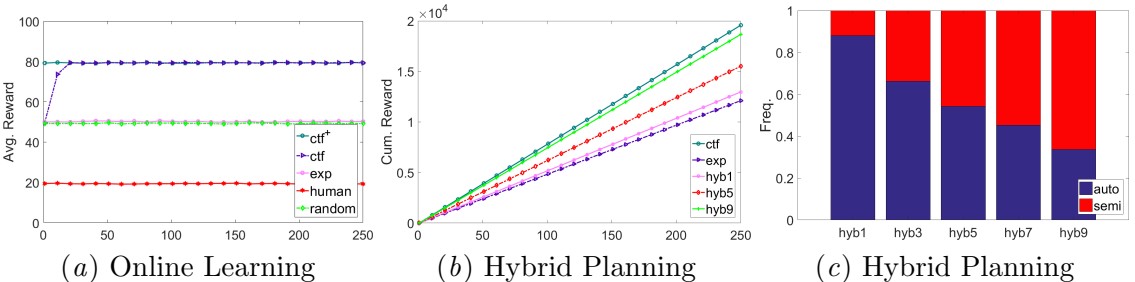

$(a)$ Online Learning $\qquad$ $(b)$ Hybrid Planning $\qquad$ $(c)$ Hybrid Planning

Figure 3: (a) Simulations comparing performance of experimental ($A_{\mathrm{EXP}}$) and counterfactual ($A_{\mathrm{CTF}}$) agents in online learning settings; (b-c) Simulations comparing the performance (b) and composition (c) of *exp* and *ctf* modes of hybrid agents. $X$-axis represents the total episodes, and $Y$-axis in (c) represents the ratio of total time for agents running in the *exp* or *ctf* mode during executions.

## 6. Applications and Experiments

In this section, we operationalize the learning algorithms for experimental ($A_{\mathrm{EXP}}$), counterfactual ($A_{\mathrm{CTF}}$), and hybrid ($A_{\mathrm{HYB}}$) agents in the context of the medical treatment problem described in Sec. 2. We assess agents' performance using the cumulative reward (CR) and the average reward (AR). Overall, we find that simulation results confirm that the counterfactual approach achieves a higher expected return, even when human decisions are poor; the polynomial program formulation effectively balances between autonomy and optimality, subject to a budget constraint over available human input.

**Experiment 1: Offline Planning.** Since $M_{\mathrm{MDPUC}}$ is both exp-Markov and ctf-Markov (Thm. 10), the best possible policies of $A_{\mathrm{EXP}}$ and $A_{\mathrm{CTF}}$ can be computed using standard MDP algorithms following the construction given in Prop. 11, labeled as *exp* and *ctf*. The performance of agents $A_{\mathrm{EXP}}$ and $A_{\mathrm{CTF}}$ are compared with the baseline policies described in Sec. 2 (*human* and *random*). An experimental policy computed through POMDP algorithms (called *exp2*) is also included. Simulations, shown in Fig. 1, support the efficiency of the counterfactual approach, revealing that (1) both experimental policies obtained by standard MDP (*exp*, CR $= 1.2162 \times 10^4$) and POMDP (*exp2*, CR $= 1.2164 \times 10^4$) coincide with *random* policy (CR $= 1.2147 \times 10^4$); (2) the counterfactual agent $A_{\mathrm{CTF}}$ (*ctf*, CR $= 1.965 \times 10^4$) consistently outperforms the experimental agent $A_{\mathrm{EXP}}$, even when the human performs worse than random guessing (*human*, CR $= 4.6492 \times 10^3$).

**Experiment 2: Online Learning.** We next show that the challenge considered here is not only bounded to offline-planning, but appears compounded in the online learning setting when the agent has only an incomplete model of the environment. We apply Mormax (Szita and Szepesvári, 2010) for experimental and counterfactual agents since $M_{\mathrm{MDPUC}}$ is both exp-Markov and ctf-Markov. To empirically estimate system dynamics of $A_{\mathrm{CTF}}$ (Prop. 11), we follow the counterfactual randomization procedure in (Bareinboim et al., 2015).

We collect trajectories for both experimental and counterfactual agents, respectively, *exp* and *ctf*. We also deploy a counterfactual agent $A_{\mathrm{CTF}}$ where the transition function $T$ and the reward function $R$ are warm-started with 5000 observational samples (labeled as

$ctf^+$, following the *composition* axiom (Pearl, 2000, pp. 229). That is, for any $x, s, x', s', x''$,

$$T_x^{\langle s,x \rangle}\left(s', x''\right) = P\left(S^{(t+1)} = s', X^{(t+1)} = x'' \mid S^{(t)} = s, X^{(t)} = x\right), \tag{12}$$

$$R_x^{\langle s,x \rangle} = E\left[Y^{(t)} \mid S^{(t)} = s, X^{(t)} = x\right]. \tag{13}$$

The two baseline policies previously described, *obs* and *random*, are also included. The simulations (Fig. $3(a)$) support the counterfactual approach. In fact, there is a significant difference in rewards experienced by counterfactual agents (AR = 79.2438 for $ctf^+$, AR = 79.2837 for *ctf*) when compared to the experimental agent *exp* (AR = 50.1900), despite the fact that *human* (AR = 19.1913) performs poorly, even worse than random guessing (*random*, AR = 49.2212). In addition, $ctf^+$ demonstrates better convergence rate (after 2 episodes) compared to *ctf* (does not converge until 25 episodes).

**Experiment 3: Hybrid Planning.** We apply the methods discussed in Sec. 5 to obtain the optimal policy for hybrid agents $A_{\mathrm{HYB}}$ in the constrained TOC setting. Recall that $\delta$ is a constraint over the ratio between the total time of running the counterfactual approach (using human input) and the total running time of the system. We compute policies for three hybrid agents with the ratio constraint $\delta$ set to $0.1, 0.5, 0.9$, labeled as *hyb1*, *hyb5* and *hyb9*, respectively. The two baseline policies *exp* and *ctf* are also included for comparison. The simulations (Fig. $3(b)$) reveal that the performance of hybrid agents converges to the counterfactual agent as $\delta \to 1$. In particular, *hyb1* ($\delta = 0.1$) shows limited performance improvement (CR = $1.3023 \times 10^4$) over the autonomous approach (*exp*, CR = $1.2162 \times 10^4$), while *hyb9* ($\delta = 0.9$) experiences higher rewards (CR = $1.8736 \times 10^4$), which is comparable to the counterfactual agent *ctf* (CR = $1.965 \times 10^4$). Predictably, the performance of *hyb5* ($\delta = 0.5$, CR = $1.558 \times 10^4$) lies in between *hyb1* and *hyb9*. We show the composition graph of experimental and counterfactual modes in Fig. $3(c)$; we also includes two hybrid agents with $\delta = 0.3, 0.7$. The simulations support that the polynomial optimization reduction worked as expected, where *hyb1* ($P(a_0) = P(A^{(t)} = a_0) = 0.88$) and *hyb3* ($P(a_0) = 0.6629$) tend to stay in the autonomous mode, while *hyb7* ($P(a_0) = 0.3366$) and *hyb9* ($P(a_0) = 0.1239$) ask for human involvement frequently; unsurprisingly, *hyb5* kept neutral.

## 7. Conclusions

We investigated a novel interactive reinforcement learning setting where the human instructor and the RL agent do not perceive the world in the same fashion, but share the reward and transition functions. We showed that the agent could be constrained to find a sub-optimal policy if it does not take the human into the loop. We proposed a new type of interactive agent, which could account for the human in real-time using counterfactual reasoning. Our analysis revealed that the counterfactual approach dominates standard methods even if the human instructor performs poorly, possibly worse than random guessing. In fact, the human should be kept "in the loop" as long as it has access to information about the tasks at hand, even after the agent completes its learning and builds a model of the environment. To resolve the tension between the autonomy and optimality of the system, we proposed a novel RL task subject to a budget constraint. Automated decision-making systems are playing an increasingly prominent role in society, and we hope this work constitutes a step towards a better understanding of the principles underlying human-machine interactions.

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
