# OpenReview forum: "Can Humans Be out of the Loop?"
_cclear.cc/CLeaR/2022/Conference — CLeaR 2022 Poster_

### Official Review · Reviewer_FQqZ · 2021-11-21

**Confidence:** 3
**Overall Score:** 7

**Main Review:**

Strengths:

The theory developed herein is relatively self-complete. The simulation studies are relatively comprehensive. The polynomial optimization formulation of the hybrid agent is quite novel and potentially has strong practical and theoretical relevance (from the algorithmic perspective). This may also open up the possibility of studying statistical-computational trade-offs in settings related to this paper. In particular, the hybrid setting is also related to recent literature on combining data with no unmeasured confounding with observational data with unmeasured confounding to dynamical settings. In this case, though, the trade-off between causal and non-causal can be balanced under the resource (access to human decision-makers) constraint.

Weaknesses/Other comments:

1. The paper did not reference many important works on time-varying causal inference (which is essentially RL + causality) in statistics: e.g. Murphy JRSSB 2003, Robins 2004, Dudík-Erhan-Langford-Li Stat. Sci. 2014.

2. From the perspective of causality, it is not at all surprising that the Experimental Agents (EA) as defined in this paper may learn sub-optimal policies when EA does not have access to all confounders. I believe this should be discussed in detail in the first section and the intuition by causal thinking should also be presented. I think it is also important to explain the intuition to readers why even when human decision-makers are inferior to random guessing, it is still beneficial to have this component in the algorithm.

3. As I mentioned, the hybrid setting seems to be closely related to the recent literature in causality that tries to combine real-world data for which strong ignorability fails to hold and data with no unmeasured confounding. Could the authors briefly comment on this?

4. In the hybrid setting, which algorithm is used for the polynomial optimization problem? How long does it take to run the algorithm? I had the impression that Sherali-Adams is fast but SDP is generally not scalable to even moderately big systems. It might be helpful if the authors can further comment on the practicality of these choices, in particular during the presentation of the simulation results.

5. MDP might not be a realistic assumption in particular in clinical applications. It might be helpful if the authors could make some further comments on how to extend their works to non-MDP settings in the Discussion section.

Overall, the flavor of causality in this paper is not very strong and the kind of idea borrowed from causality is rudimentary. But the optimization algorithms, the theoretical justification, and the detailedness of the authors' analyses are worth sharing with the causal inference community.

Minor typos:

Corollary 19 proof (Appendix page 17): $F_{CTF}$ should be $\Pi_{CTF}$.

**Summary:**

This paper considers the following setting of RL: one agent can only access part of the confounders in the system. Even this agent can access all the reward function and state-transition functions, the learnt policy may still be sub-optimal. Then the authors propose the so-called "counterfactual agent" and a hybrid approach to improve the regret.

---

> ### Author Response · Authors · 2021-12-04
> **Response to Reviewer FQqZ (i)**
>
> We thank the reviewer for the thoughtful feedback and incorporate it into the updated manuscript. We summarize the reviewer’s questions and provide answers below
>
> ---
> #### Q1 + Q3
> >"The paper did not reference many important works on time-varying causal inference (which is essentially RL + causality) in statistics: e.g. Murphy JRSSB 2003, Robins 2004, Dudík-Erhan-Langford-Li Stat. Sci. 2014.”
>
> >“As I mentioned, the hybrid setting seems to be closely related to the recent literature in causality that tries to combine real-world data for which strong ignorability fails to hold and data with no unmeasured confounding. Could the authors briefly comment on this?”
>
> Thank you for sharing the references. The problem settings the reviewer mentioned are related, but orthogonal to the one studied in our work. More specifically, these papers studied the offline learning setting where the goal is to learn an efficient interventional policy from data collected through observational studies. The learned policy is still “experimental” (Definition 3), i.e., it does not utilize the human’s action during the execution of the policy. In other words, the human’s action is only utilized before the deployment of the policy and is ignored after a policy is obtained. There is a good amount of follow-up on this work, which includes graphical models, for example, see (Zhang and Bareinboim 2017, 2019, 2020, 2021).
>
> On the other hand, this paper studies another aspect of issues arising due to unmeasured confounders. We design a novel family of agents that models the system dynamics using a specific type of counterfactual distributions in Definition 8, similar to the effects of the treatment on the treated (ETT). Using this counterfactual representation, the agent is able to actively account for the human’s action “on the fly”, after the deployment of the policy.  (Here, interventional and counterfactual policies are strictly different, as shown in Def. 3 and Def. 5. )
>
> Finally, we note that there exist opportunities combining both the offline and online modes of learning. For instance, in Experiment 2, we learn some entries of ETT distributions from the confounded observational distribution (Eqs. 12-13). The agent then uses the estimated ETT probabilities to learn a counterfactual policy that actively accounts for the human’s action.
>
> ---
> #### Q2 “From the perspective of causality, it is not at all surprising that the Experimental Agents (EA) as defined in this paper may learn sub-optimal policies when EA does not have access to all confounders. I believe this should be discussed in detail in the first section and the intuition by causal thinking should also be presented. I think it is also important to explain the intuition to readers why even when human decision-makers are inferior to random guessing, it is still beneficial to have this component in the algorithm.”
>
> We agree with the reviewer that experimental agents “may learn sub-optimal policies when EA does not have access to all confounders.” Humans’ suboptimal decisions are still beneficial since the agent may use them to infer the optimal decision. For instance, for a binary action X, if an intervention X = 0 is inferior to random guessing, the opposite action X = 1 is guaranteed to perform better than random guessing. We will include further explanations in the updated manuscript. Thank you for the suggestion.
>
> ---
> #### Q4 “In the hybrid setting, which algorithm is used for the polynomial optimization problem? How long does it take to run the algorithm? I had the impression that Sherali-Adams is fast but SDP is generally not scalable to even moderately big systems. It might be helpful if the authors can further comment on the practicality of these choices, in particular during the presentation of the simulation results.”
>
> We used the SDP relaxation (Lasserre, 2001) to solve the polynomial program for all experiments. It could return an optimal hybrid policy in 10-15 mins. However, the program was unable to finish for a moderately large domain due to the lack of memory. We conducted experiments on a laptop with 32GB memories. Similar to standard RL algorithms, our method could face computational challenges when the state-action space is high-dimensional. To address this issue, one approach is to consider additional parametric assumptions in the transition distribution and the reward function, e.g., see (Guestrin et al., 2003).

---

> ### Author Response · Authors · 2021-12-04
> **Response to Reviewer FQqZ (ii)**
>
> ---
> #### Q5 “MDP might not be a realistic assumption in particular in clinical applications. It might be helpful if the authors could make some further comments on how to extend their works to non-MDP settings in the Discussion section.”
>
> This paper studies the problem of maximizing the cumulative discounted reward over an infinite horizon ($T = \infty$). In clinical applications, most environmental models (e.g., dynamic treatment regimes (DTRs)) are concerned with the finite-horizon decision-making settings. For instance, every patient usually only receives a finite number (say, 10) of chemotherapy treatments. It follows immediately that one could define counterfactual agents for DTR models that actively account for the physician’s action. However, the polynomial program formulation for the trade-off between autonomy and optimality depends on the MDP properties and may require additional generalization. We will include further discussions about extending to non-Markov settings in the updated manuscript.

---

> > ### Comment · Reviewer_FQqZ · 2021-12-26
> > **thank you**
> >
> > I'd like to thank the authors for their comprehensive response. The responses have fully addressed my concerns. I'll increase my evaluation by one point.
> >
> > Thank you!

---

### Official Review · Reviewer_yKxJ · 2021-11-23

**Confidence:** 3
**Overall Score:** 6

**Main Review:**

Significance:
This paper studies an interesting topic, i.e., what is the role of humans for interactive reinforcement learning settings. To formally study this, the paper defines the experimental agent and counterfactual agent and identifies the conditions that when human input is valuable to achieve optimal performance. The interesting part is that it shows when human has specific information for the task that is not accessible to the agent, though they do not perform it optimally, it could still boost the performance of the agent.

Quality/Clarity:
This paper is easy to read. By formally defining the related concepts in the causal language, the setup is clear and makes the paper easy to follow and understand. I appreciate the identification of the conditions that when human input can provide valuable information. It would be great (if possible) to characterize this in a more quantitative way. Intuitively the quality of human play should have some effect on the learning process of the agent, this might could be achieved by imposing some human players' policy and seeing how it affects the final performance. Another interesting part I like of the paper is the trade-off between optimality and autosomally through a polynomial optimization problem, though it would be great if the paper could discuss some scalability of the optimization problem.

Minor comments:
Maybe I am missing something here, the paper mentions that the agents could still benefit from the human's advice though the reward and transition are fully known (with some features unobservable), is there any experiments showing this empirically?



**Summary:**

This paper studies the role of human instructor in the interactive setting, and shows that it is helpful to keep human in the loop as long as it has information regarding to the task, though not playing it in an optimal way.

---

> ### Author Response · Authors · 2021-12-04
> **Response to Reviewer yKxJ**
>
> We thank the reviewer for the thoughtful feedback. We agree that a quantitative condition for characterizing the value of the human’s action is an interesting research direction. We summarize the reviewer’s questions and provide answers below.
>
> ---
> #### Q1. “Another interesting part I like of the paper is the trade-off between optimality and autosomally through a polynomial optimization problem, though it would be great if the paper could discuss some scalability of the optimization problem.”
>
> We are glad that the reviewer appreciated the trade-off between optimality and autonomy. Our polynomial optimization problem faces the same challenges as other standard MDP planning algorithms with high-dimensional state space. One approach to addressing these challenges is exploring additional parametric assumptions about the transition distributions and the reward functions, e.g., the linearity.
>
> ---
> #### Q2. “Maybe I am missing something here, the paper mentions that the agents could still benefit from the human's advice though the reward and transition are fully known (with some features unobservable), is there any experiments showing this empirically?”
>
> Yes, Figure 1 shows such an experiment. More specifically, exp2 is a policy obtained from standard POMDP planning algorithms provided with the full knowledge of the transition distribution $P(s^{(t+1)}|s^{(t)}, u^{(t)}, x^{(t)})$ and the reward function $P(y^{(t)}|s^{(t)}, u^{(t)}, x^{(t)})$. It uses $U^{(t)}$ as the latent state and $S^{(t)}$ as its partial observation. The ctf represents the counterfactual policy learned using human input. Simulation results in Figure 1 reveal that the performance of the standard POMDP agent (exp2) coincides with the random coin-flipping (random). The counterfactual agent (ctf) outperforms all the other policies, even though the physician (human) performs worse than random guessing.

---

### Official Review · Reviewer_DkQi · 2021-11-24

**Confidence:** 3
**Overall Score:** 6

**Main Review:**

The paper is generally well written and present interesting ideas within the specific MDPUC and DSCM models. My major concern is whether the foundation results, namely, Cor 6 and Thm 7, are indeed meaningful.
-	First, it is assumed that agent cannot access $U$, but human can. My question is why this is a rational setting?
-	Second, in Thm 7, the condition “the human decision is affected only by observable variables” indicates that the human decisions cannot bring in more information than what the agent observes. In other words, as no more information is introduced, it is intuitive that human decisions are not useful if we consider the asymptotic performance.
-	Third, I do not find any condition on the human decision, which is a bit surprising to me. E.g., even if human can access $U$, but his/her decision is a constant (say, always 1), then it does not provide any information.

Minor: $U^{(t1)}, U^{(t2)}$ seems not defined.

In summary, the paper presents interesting ideas, with the specific RL setting. However, it seems to me that whether the human can help is equivalent to whether there could be more information. In this sense, the contribution of the paper is weakened.


**Summary:**

This paper considers an interactive RL ting where the agent and the human may have different sensory capacities. It then shows agents may learn only sub-optimal policies if they do not take into account human advice. A counterfactual agent is proposed to include the results of actions from humans. A new RL algorithm is also proposed considering that there is a budget constraint over the available amount of human advice.

---

> ### Author Response · Authors · 2021-12-04
> **A few clarifications**
>
> We thank the reviewer for the thoughtful feedback. We believe that a few misreadings of our work made some of the evaluations overly harsh. We respectively ask the reviewer to reconsider our paper in the light of clarifications provided below.
>
> ---
> #### Q1. “First, it is assumed that agent cannot access $U$, but human can. My question is why this is a rational setting?”
>
> Yes, we believe that this setting represents many challenges in real-world applications. For instance, most vehicles with “autonomous driving” features today are indeed “semi-autonomous”, i.e., they require a human operator at the driver seat. The human driver monitors the road condition and could take over the control when the situation becomes complicated. Since the human driver and the robot driver (i.e., the vehicle) have different sensory capabilities, the robot might not necessarily be able to obtain some information accessible to the human, i.e., they are unobserved confounders to the robot. For instance, the human driver could hear the siren and decide to stop the car on the sideway. However, sound information is not recorded in most autonomous driving datasets (e.g., ApolloScape, 2018). The omission of audio information could be  In this case, the siren becomes a confounder unobserved to the robot driver. If the robot decides to ignore the human’s action (i.e., stopping the car), it may lead to accidents and even causalities.
>
> ---
> #### Q2. “Second, in Thm 7, the condition “the human decision is affected only by observable variables” indicates that the human decisions cannot bring in more information than what the agent observes. In other words, as no more information is introduced, it is intuitive that human decisions are not useful if we consider the asymptotic performance.” ... “However, it seems to me that whether the human can help is equivalent to whether there could be more information. In this sense, the contribution of the paper is weakened.”
>
> We are glad that the reviewer found our graphical condition in Theorem 7 “intuitive”. While it may appear straightforward, Theorem 7 is the first condition in the literature that determines whether full autonomy could be achieved in automated systems, which we believe is a significant contribution. Also, Theorem 7 only constitutes part of our contributions. We provide in Section 5 a novel method to resolve the trade-off between autonomy and optimality. It finds an effective semi-autonomous decision policy that optimizes the reward while subject to budget constraints over humans’ involvement.
>
> ---
> #### Q3. “Third, I do not find any condition on the human decision, which is a bit surprising to me. E.g., even if human can access $U$, but his/her decision is a constant (say, always 1), then it does not provide any information.”
>
> Theorem 7 states that the human’s action could be ignored if “the human decision is affected only by observable variables.” The reviewer’s example, i.e., the human’s decision “is a constant (say, always 1)” regardless of values of $U$, satisfies this condition.
>
> ---
> #### Q4. “$U^{(t1)}, U^{(t2)}$ is seems not defined.”
>
> In the MDPUC+ of Figure 2(c), the unobserved states $U^{(t)} = (U^{(t1)}, U^{(t2)})$” (Page 6, Line 6). That is, $U^{(t1)}, U^{(t2)}$ are elements in the random vector $U^{(t)}$.

---

> > ### Comment · Reviewer_DkQi · 2021-12-13
> > **Thanks for response**
> >
> > Thanks for your response. I decide to slightly increase my evaluation to "6: Marginally above acceptance threshold". (but it seems I cannot modify the original comment. I will figure it out.)
> >
> > The given example is a good motivating example, so please make it clear in the paper. And also maybe add some more realistic experiments, which should benefit the paper. I

---

### Decision · Program_Chairs · 2022-01-12

**Decision:**

Accept (Poster)

**Comment:**

Thank you for your submission. Reviewers appreciated this paper's use of counterfactual reasoning to analyze and characterize when human advice (including poor advice) based on state unobserved by an agent can inform a more optimal dynamic policy. The paper's setting is interesting, and the work provides meaningful results. Reviewers particularly appreciated the papers' formulation of the optimization problem and the trade-off between optimization and autonomy.

We appreciate the motivating example and intuition provided by the authors in their response; and encourage them to include this in a final revision, together with other details and clarifications from the author responses to reviewer questions.